# The Ambivalent Role of Skin Microbiota and Adrenaline in Wound Healing and the Interplay between Them

**DOI:** 10.3390/ijms22094996

**Published:** 2021-05-08

**Authors:** Arif Luqman, Friedrich Götz

**Affiliations:** 1Biology Department, Institut Teknologi Sepuluh Nopember, Surabaya 60111, Indonesia; 2Microbial Genetics, Interfaculty Institute of Microbiology and Infection Medicine Tübingen (IMIT), University of Tübingen, D-72076 Tübingen, Germany

**Keywords:** skin commensal, trace amines, adrenaline, adrenergic receptors, wound healing, health

## Abstract

After skin injury, wound healing sets into motion a dynamic process to repair and replace devitalized tissues. The healing process can be divided into four overlapping phases: hemostasis, inflammation, proliferation, and maturation. Skin microbiota has been reported to participate in orchestrating the wound healing both in negative and positive ways. Many studies reported that skin microbiota can impose negative and positive effects on the wound. Recent findings have shown that many bacterial species on human skin are able to convert aromatic amino acids into so-called trace amines (TAs) and convert corresponding precursors into dopamine and serotonin, which are all released into the environment. As a stress reaction, wounded epithelial cells release the hormone adrenaline (epinephrine), which activates the β2-adrenergic receptor (β2-AR), impairing the migration ability of keratinocytes and thus re-epithelization. This is where TAs come into play, as they act as antagonists of β2-AR and thus attenuate the effects of adrenaline. The result is that not only TAs but also TA-producing skin bacteria accelerate wound healing. Adrenergic receptors (ARs) play a key role in many physiological and disease-related processes and are expressed in numerous cell types. In this review, we describe the role of ARs in relation to wound healing in keratinocytes, immune cells, fibroblasts, and blood vessels and the possible role of the skin microbiota in wound healing.

## 1. The Ambivalent Role of the Skin Microbiota in Wound Healing

It is textbook knowledge that the wound healing process occurs in four stages [1,2]: hemostasis, inflammation, proliferation, and maturation [3,4]. Hemostasis is the process of wound closure by clotting. The inflammatory phase begins immediately after the injury, when transudate (consisting of water, salt, and protein) leaks from the injured blood vessels and causes local swelling. The inflammation has two functions: it controls bleeding and counteracts infection. In the proliferative phase, the wound is built up with new tissue from collagen and extracellular matrix. The wound contracts with the help of myofibroblasts that grip the wound edges and pull them together with a mechanism similar to that of smooth muscle cells. Furthermore, a new network of blood vessels is established (angiogenesis) so that the granulation tissue is healthy and receives sufficient oxygen and nutrients [5]. The maturation phase is the remodeling phase of wound healing in which collagen is converted from type III to type I and the wound closes completely. The cells that were used to repair the wound but are no longer needed are removed by apoptosis, or programmed cell death.

Normally, the phases of wound healing are linear; however, wounds can progress backward or forward depending on internal and external patient conditions. Failure to progress in the stages of wound healing can lead to chronic wounds. Chronic wounds frequently occur in patients with underlying disorders such as venous or arterial insufficiency, infection, diabetes, immunosuppression, poor nutrition, cell hypoxia, dehydration, or metabolic deficiencies of the elderly [6]. The transition from the inflammatory to the proliferative phase is a key step during healing, and a compromised transition is associated with wound healing disorders that can lead to chronic or nonhealing wounds [7]. The nonhealing wounds are often complicated by bacterial infections that trigger a continuous influx of neutrophils and macrophages, further delaying wound healing [8]. However, many studies have also reported the positive effect of the skin microbiota in wound healing, either by modulating immune response, enhancing the wound healing process, or preventing pathogen infection.

## 2. Certain Members of the Skin Microbiota Can Have a Negative Impact on Wound Healing

The skin represents the primary interface between the host and the environment. This organ is also home to trillions of microorganisms, commensal microbiota, that play an important role in tissue homeostasis and maintaining a symbiotic relationship with the immune system [9,10]. The commensals can be remodeled over time or in response to environmental challenges. The precise relationship between the commensal microbiota and impaired wound healing remains unclear. In a diabetic mouse model, it was shown that over time the spectrum of colonizing bacteria shifted to Firmicutes species (including *Staphylococcus* and *Aerococcus*), which was correlated with enhanced expression of genes involved in defense and immune response [11]. Moreover, *Corynebacterium striatum*, *Alcaligenes faecalis*, and *S. aureus* have a negative impact on diabetic wound healing and severity [12], and biofilm-forming bacteria such as *S**. aureus, Pseudomonas aeruginosa, Peptoniphilus, Enterobacter, Citrobacter freundii, Escherichia coli, Klebsiella pneumoniae, Proteus mirabilis, Stenotrophomonas, Finegoldia,* and *Serratia* spp. impair diabetic wound healing [13,14,15].

In a study of diabetic foot ulcer (DFU), it was shown that the proportions of *Bacteroidetes, Prevotella, Peptoniphilus, Porphyromonas*, and *Dialister* were higher in the severe groups than in the mild groups, whereas that of Firmicutes was lower in the severe groups [16]. In refractory chronic venous leg ulcers, anaerobic bacteria, particularly *Peptostreptococcus* spp., were present in the deep tissues, where they inhibit keratinocyte wound repopulation and endothelial tubule formation [17]. Moreover, in patients with neuropathic DFU, it was shown that the most abundant genera in a study with neuropathic DFU were *Staphylococcus* (18%), *Corynebacterium* (14%), *Pseudomonas* (9%), and *Streptococcus* (7%) [12]. In this study *S. aureus* led to a deterioration of wound healing, whereas the typical commensals *C. striatum* and *A. faecalis* enhanced wound re-epithelialization.

*S. aureus* generally appears to play a negative role in wound healing. Chronic wounds are characterized by high levels of oxidative stress (OS). To verify whether OS really impairs wound healing, catalase inhibitors were applied in a diabetic mouse wound model, and it was shown that increasing OS levels were correlated with increasing chronicity [18]. Interestingly, high OS levels in the wound tissue in the absence of the skin microbiome do not lead to chronic wounds. These findings show that both high OS levels and bacteria are needed for chronic wound initiation and development. *S. aureus*, which is a commensal on the healthy human skin [19], is frequently reported to play a negative role in wound healing. However, only a few studies differentiate whether they are Agr+ or Agr− negative strains. Agr is a global regulator that particularly controls virulence gene expression. Agr− strains hardly produce hemolysins and phenol-soluble modulins (PSMs). Particularly, the N-formyl methionine-containing (fMet) PSMs not only have a high cytotoxic activity but also activate the FPR2 receptor, leading to neutrophil activation, chemotaxis, and cytokine release [20,21].

Regardless of the distinction between friend and foe in the skin microbiota, it can be said that overall, it tends to play a negative role. This is supported by the observation that wound repair of germ-free (GF) mice was much better compared to conventional (CV) mice with a commensal microbiota [22]. In GF mice, levels of the anti-inflammatory cytokine IL-10, the angiogenic growth factor VEGF, and angiogenesis were higher. Overall, this study suggests that in the absence of any contact with microbiota, skin wound healing is accelerated and scarless, partially because of reduced accumulation of neutrophils, increased accumulation of alternatively activated healing macrophages, and better angiogenesis at wound sites [22]. That the microbiota is a primary cause for the pathogenesis of chronic wounds was impressively demonstrated by seeding the microbiota from a human chronic wound to a mouse that developed a similar chronic wound [23].

## 3. Positive Effect of Some Commensals and Probiotics on Wound Healing

The skin microbiota is an ecosystem comprising a multitude of microbial species interacting with their surroundings, including other microbes and host epithelial and immune cells [24]. We know that the skin microbiota is in principle good if it is in balance with the skin immune system and protects the skin from pathogens. We also know that the skin microbiota can be remodeled over time to a predominantly ‘bad’ population. However, it is not so easy to determine which are the good and which are the bad bugs.

It is assumed that excessive and, above all, permanent stimulation of the innate immune system and inflammatory responses of the host not only worsens wound healing but also promotes a chronic course [25,26,27]. Apparently, commensal and pathogenic bacteria differ in their immune stimulation and expression of defensins. For example, commensal staphylococci induce antimicrobial peptides/proteins (AMPs) via TLR2, EGFR, and NF-κB activation, whereas pathogenic staphylococci activate the mitogen-activated protein kinase and phosphatidylinositol 3-kinase/AKT signaling pathways and suppress NF-κB activation [28].

Skin wounds heal by coordinated induction of inflammation and tissue repair; thereby, commensal skin microbiota can play a positive role by the activation of type I interferon (IFN)-producing plasmacytoid DC (pDC) [29]. This activation leads to the expression of the chemokine CXCL10, which recruits pDC and acts as an antimicrobial protein to kill exposed microbiota, leading to the formation of CXCL10–bacterial DNA complexes. The bacterial DNA complexes and not the host-derived DNA activate pDC to produce type I IFNs, which accelerate wound closure by triggering skin inflammation and early T-cell-independent wound repair responses, mediated by macrophages and fibroblasts that produce major growth factors required for healing [29]. Besides, particular commensals, such as *S. epidermidis*, were reported to induce IL-17A(+) CD8(+) T cells that home to the epidermis, enhance innate barrier immunity, and limit pathogen invasion [9].

Although *S. aureus* is the most prominent cause of skin and soft tissue infections (SSTI) worldwide, it has been shown that the two IgG-binding proteins, SpA and Sbi, play a positive role in skin repair and wound healing, because corresponding mutants show an increased abscess formation, increased bacterial load in skin lesion, and increased skin lesion area [30]. However, what has not been shown is how the direct treatment with SpA or Sbi affects wound healing.

Some coagulase-negative staphylococci (CoNS) species can inhibit quorum sensing (QS) or even the growth of *S. aureus*. *Staphylococcus caprae*, for example, produces the *S. caprae* autoinducing peptide (AIP) that inhibits QS of *S. aureus* and thus the expression of various virulence genes [31]. *Staphylococcus lugdunensis* strains produce lugdunin, a novel thiazolidine-containing cyclic peptide antibiotic that prohibits colonization by *S. aureus* [32]. Quite a number of CoNS produce 6-thioguanine (6-TG), a purine analog that suppresses *S. aureus* growth by inhibiting de novo purine biosynthesis and is also effective in wound healing [33]. Other CoNS produce lantibiotics such as epidermin and gallidermin, which inhibit cell wall biosynthesis and have broad-spectrum activity against Gram-positive pathogens [34,35], or antimicrobial peptides (AMPs) that selectively kill *S. aureus* and synergize with the human AMP LL-37 [36]. All these species could have a positive influence on wound healing or prevent a chronic course. Other probiotic candidates with a positive effect are lactic acid bacteria such as *Lactobacillus plantarum, Lactobacillus casei, Lactobacillus acidophilus,* and *Lactobacillus rhamnosus* [37]. Recombinant lactobacilli expressing the chemokine CXCL12, which is strongly chemotactic for lymphocytes, also improved wound closure in mice with hyperglycemia or peripheral ischemia, conditions associated with chronic wounds [38].

Unfortunately, the knowledge we have about the negative and positive effects of some bacteria in wound healing is still fragmentary. It is therefore important to identify the causes and elicitors of these chronic and nonhealing wounds. Often microorganisms play a role that counteracts and delays the healing process by excreting effectors such as toxins and immune stimulants that counteract and delay the healing process. Evidence is accumulating that excessive or persistent inflammatory responses triggered by bacteria are major causes of chronic and nonhealing wounds. It is therefore necessary to identify the triggers of the inflammatory reactions and find ways to neutralize their activity in order to ensure the natural course of wound healing. In Table 1, skin microbiota species that have an adverse or a promoting effect on wound healing are summarized.

## 4. Adrenergic Receptors (ARs) Are Expressed in Many Cells of the Skin and Contribute to Wound Healing

A stress reaction triggered by a wound leads to an increase, both locally and systemically, in the production of stress hormones such as adrenaline and cortisol [55,56,57,58]. These stress hormones, particularly adrenaline (epinephrine), are crucial at the first stage of wound healing, as well as for the homeostasis and inflammatory phase. However, the prolonged increased level of adrenaline delays further stages of wound healing (Figure 1A), namely the proliferative and maturation phases [59]. As adrenaline plays an important role in wound healing [44], adrenergic receptors (ARs) also come into play.

ARs are a class of G protein-coupled receptors that interact with catecholamines such as noradrenaline (norepinephrine) and adrenaline. ARs are classified as alpha or beta receptors, which can be further subdivided into α-1, α-2, ß-1, ß-2, and ß-3 and even to further subtypes [60]. ARs play a key role in many physiological or disease-related processes and are present in numerous cell types involved in wound healing processes. For example, keratinocytes express β2- and α1-AR [61,62,63]; endothelial cells express β2-AR [64,65]; immune cells, such as neutrophils and macrophages, express β2-AR [66,67]; and fibroblasts express β1-, β2-, β3-, and α1-AR [68,69,70,71,72,73].

### 4.1. Keratinocytes

Keratinocytes make up over 90% of the cells of the epidermis, the outermost layer of the skin. Keratinocytes undergo a differentiation process starting from the basal membrane up to the keratinized epithelium (the outermost layer). Keratinocytes represent the first cell layer to respond to stressors such as injury or invasion of pathogens by releasing cytokines or hormones like adrenaline. They express two key enzymes for adrenaline synthesis, namely tyrosine hydroxylase and phenylethanolamine-N-methyl transferase, within cytoplasmic vesicles [74]. An injury induces an increase in adrenaline level locally by upregulating the phenylethanolamine-N-methyltransferase in keratinocytes [57]. The increase in adrenaline level is followed by the adrenaline’s exocrine and paracrine effects in the wound area. Due to their re-epithelialization capacity, keratinocytes play an important role in wound healing; they proliferate and migrate to the injured site. The activation of β2-AR in keratinocytes induces an increase in the intracellular Ca++ level [75]; thus, the keratinocyte proliferation is increased as intracellular Ca++ regulates cell proliferation [76,77].

However, the β2-adrenergic receptor activation also impairs the re-epithelization due to stabilization of the actin cytoskeleton [78], which leads to a reduction in the migration ability of keratinocytes [57]. The directional migration of keratinocytes is modulated by β2-AR activation by two distinct mechanisms: cAMP-independent and cAMP-dependent mechanisms [79,80]. In keratinocytes, β2-AR activation is followed by the increase in β2-AR association with protein phosphatase 2A (PP2A) and association of PP2A with extracellular signal-regulated kinase 2 (ERK) [79]. Moreover, β2-AR activation in keratinocytes also downregulates the Akt pathway [57]. As the ERK2 pathway and the Akt pathway are promigratory signaling cascades, these phenomena lead to a nonmigratory phenotype in keratinocytes.

### 4.2. Immune Cells

The immune system orchestrates the wound healing process by modulating the secretion of cytokines, chemokines, and growth factors to promote cell-to-cell communication [27]. ARs somehow take a role in immune modulation in wound healing, as ARs are found in lymphocytes, macrophages, and neutrophils [81,82]. For example, in the inflammation stage, the increase in adrenaline due to injury activates the macrophage’s β2-AR and induces IL-6 production. The increase in IL-6 leads to neutrophil trafficking to the wound area. This event is important to eradicate any potential pathogen to prevent infection.

However, the extended β2-AR activation due to high adrenaline production in the wound area hinders the healing process [83]. The wound healing process can be impeded by not only the increased local adrenaline but also the high level of systemic adrenaline. A study conducted by Romana-Souza in 2010 [59] using rotation-stressed mice revealed the following changes: a delay in the infiltration of neutrophils and mast cells into the wound area, a delay in TNF-α expression, and recruitment of F4/80-positive macrophages. These series of events are modulated by β-AR activation [59,78]. The mechanisms through which β2-ARs signal, how β2-AR functions and is regulated by the sympathetic nervous system (SNS), and how β2-AR cross-talks with other signaling pathways activated by immune challenge is summarized by Lorton and Bellinger [84].

### 4.3. Fibroblasts

Fibroblasts engage in wound healing by providing the extracellular matrix and collagen structures to support effective wound healing and wound contraction [85]. To perform its functions, a fibroblast needs to proliferate, migrate, and differentiate, which are processes that are regulated by ARs. β1-, β2-, and β3-AR activation induces fibroblast proliferation via different pathways. β2-AR activation increases ERK 1/2 phosphorylation and thus enhances proliferation [85]. The activation of β2-AR increases migration of fibroblasts [72], but the activation of β3-AR inhibits their migration by stimulating excessive nitric oxide production and inhibiting Akt phosphorylation. After migrating and proliferating in the wound area, fibroblasts need to differentiate and produce collagen to complete their job. Studies by Romana-Souza [59,78,85] in chronically stressed mice and murine dermal fibroblast cultures showed that β-AR activation increases myofibroblast differentiation but decreases collagen production.

### 4.4. Blood Vessels

An injury with ruptured blood vessels leads to blood loss. The adrenaline produced by keratinocytes [57] shows a paracrine effect on the smooth muscle surrounding the blood vessels. The α1-AR activation of smooth muscle leads to muscle contraction, causing the constriction of blood vessels and finally reducing the blood loss [86,87]. The ruptured blood vessels in a wound will be recovered at the angiogenesis stage, which is, however, inhibited by adrenaline.

The prolonged increase in adrenaline, both locally in the wound area and systemically, impedes angiogenesis via β-AR activation [88]. The activation of β-AR hinders the endothelial migration via cAMP-dependent and protein kinase A (PKA)-independent mechanisms. It also reduces the formation of the fibroblast growth factor 2 (FGF2) and vascular endothelial growth factor A (VEGF-A), as well as proangiogenic growth factors secreted by endothelial cells and keratinocyte cells, respectively, and suppresses endothelial tubule formation, as shown in an in vitro study using cell cultures [88]. An in vivo study using a murine excisional skin wound model also showed that activation of β-AR inhibits the formation of new blood vessels in a wound [88].

## 5. Interplay between Adrenaline and Skin Bacteria

### 5.1. TA-Producing Skin Microbiota Might Accelerate Wound Healing via ARs

Many members of the genera *Staphylococcus* and *Macrococcus* are typical colonizers of the human skin [89]. Recently, it has been shown that various staphylococcal species of the human skin microbiota possess the *sadA* gene that encodes an aromatic amino acid decarboxylase (SadA) that converts aromatic amino acids into so-called trace amines (TAs): tryptamine, phenethylamine, and tyramine [90]. The TAs are more or less quantitatively excreted and can therefore interfere with host cell specific TA-receptors. TA-producing staphylococcal species are surprisingly common in the human gut and human skin microbiota [19,44,90]. TAs can interact with various ARs. They act as agonists of α2-AR [91,92] and as antagonists of β-AR [44,93] and of α1-AR [94]. Some studies used antagonists to block β-AR, which resulted in an increase in ERK phosphorylation, keratinocyte migration, and re-epithelialization and conclusively accelerated wound healing [72,95]. It has been proven that TAs accelerate keratinocyte migration via cAMP-dependent mechanisms by blocking the β2-AR in vitro [44]. However, the positive effect of TA on wound healing was also shown in a mouse model where not only topical administration of TA but even a TA-producing *Staphylococcus epidermidis* strain accelerated wound healing in contrast to its non-TA-producing mutant [44]. Since SadA is a highly promiscuous TA-producing decarboxylase in Firmicutes, the skin microbiome was specifically examined for the presence of *sadA*-homologous genes. SadA is a highly promiscuous TA-producing decarboxylase in Firmicutes, and its homologs were found in seven bacterial phyla and a large number of genera of the human skin microbiome [19]. The TA-producing species among the microbiota enhance wound healing most likely at the early hemostasis stage. These results support the hypothesis that skin commensals might play a positive role in wound healing (Figure 1B).

However, the role of TA-producing skin commensals in wound healing is probably not only due to the acceleration of keratinocyte migration. Because many crucial wound healing processes are slowed down by β-AR activation, it is quite possible that TAs produced by skin commensals also accelerate wound healing by initiating counteracting processes. We think that the infiltration of neutrophils and mast cells into the wound region, the recruitment of F4/80-positive macrophages, TNF-α expression, fibroblast proliferation–migration–differentiation, and angiogenesis processes might also be affected by TAs. To better understand the influence of the skin microbiota on wound healing, we need more knowledge on the molecular mechanisms of β-adrenergic receptor-mediated cross-talk between sympathetic neurons on the one hand and epithelial and immune cells on the other. We also need to know more about TA-induced receptor transactivation cascades.

### 5.2. Adrenaline Controls Not Only Sympathetic Nervous System but Also Quorum Sensing in Bacteria

Adrenaline also controls metabolism and virulence traits of some members of the skin microbiota. The spike in adrenaline due to injury affects not only the skin tissues themselves but also members of the skin microbiota (Figure 2). Some bacterial species that are normally found on skin, *Micrococcus luteus* and *Cutibacterium acnes,* showed an increase in biofilm formation in the presence of adrenaline [96,97]. The increased biofilm formation, particularly with *C. acnes*, is due to the possession of a catecholamine receptor homolog as found in eukaryotes and *E. coli* [96]. Catecholamine receptors in bacteria have been reported first in *E. coli* as a quorum sensing receptor named QseC that regulates some phenotypes, such as motility and adherence. Later, QseC homologs were reported in many other Gram-negative bacteria [98,99,100,101,102]. Recently, similar receptors were also discovered in Gram-positive bacteria, such as VicK in *Enterococcus faecalis* that regulates adherence to keratinocytes and biofilm formation as well [103].

Adrenaline also has been reported to support the growth of bacteria in general by forming complexes with ferric ions as a siderophore [104]. In one study, using a guinea pig model of surgical wound infection, it was shown that treatment of adrenaline together with lidocaine exhibited a 20 times higher *S. aureus* colonization in the wound compared to treatment with lidocaine alone [105]. Adrenaline also increases the metabolic activity of *Micrococcus luteus* [97] and the swarming activity of *Pseudomonas fluorescens* [106].

## 6. Topical Probiotics Are a Therapeutic Option in the Treatment of Chronic Wounds

Wound healing is a complex process and is orchestrated by a sophisticated interplay between factors. One of the intrinsic factors is adrenaline, which is crucial for the early stages of the wound healing process via AR activation. The extended AR activation, however, impedes keratinocyte migration, decreases collagen production, and delays the immune cell activities and angiogenesis process. To overcome these impediments by ARs, the administration and application of AR antagonists, particularly β-AR antagonists, showed promising results both in in vitro and in vivo studies. Studies by Romana-Souza [59,78] in stressed mice revealed that the administration of propranolol, a β-AR antagonist, reversed the delay in infiltration of neutrophils and mast cells into the wound area, a delay in TNF-α expression, and recruitment of F4/80-positive macrophages, which was later followed by faster wound closing and re-epithelization. Besides propranolol, dopamine and TA application could increase the keratinocyte migration rate by inhibiting the increase in cAMP level through partial blockade of β2-AR and eventually accelerate wound healing [44].

Another potential therapeutic strategy to enhance wound healing is via skin microbiota manipulation. The purpose of skin microbiota manipulation is to increase the number of advantageous microbiota, particularly in the wound healing process, either by preventing pathogen infection or by accelerating the healing process (Table 1). As mentioned above, TAs can be produced by skin commensals that possess sadA gene. Indeed, it has already been demonstrated that topical application of TA-producing bacteria (*S. epidermidis* O47) on skin could enhance wound healing [44]. Even certain lactobacilli, which are not typical skin commensals, and their lysate can increase keratinocyte migration and proliferation [49]. Moreover, lactobacilli are known to produce organic acids that show antimicrobial activity against skin pathogens and inhibit biofilm formation on wounds [37,47,48]. Skin microbiota manipulation can be also carried out by topical application of certain compounds, such as traditional Chinese medicine that has been proven to be effective in treating chronic ulcers by regulating wound microbiota [107].

## 7. Conclusions

Skin microbiota and ARs play an ambivalent role in wound healing. On the one hand, they can hinder the wound healing process and lead to chronic wounds; on the other hand, some bacterial species can accelerate the wound healing process and suppress the colonization of pathogens. Skin injury induces the stress hormone adrenaline, which delays early-stage wound healing by activating ß-ARs in skin cells. However, adrenaline not only affects skin cells but also promotes virulence and growth of unwanted bacteria. TA-producing skin commensals can override the effect of adrenaline and thus positively influence wound healing (Figure 3). These bacteria are part of the skin microbiota. However, their proportion varies from person to person and may be so low for some that the positive effect does not come into play. Therefore, selected TA-producing commensal bacteria may represent a promising therapeutic option in wound healing.

## Figures and Tables

**Figure 1 ijms-22-04996-f001:**
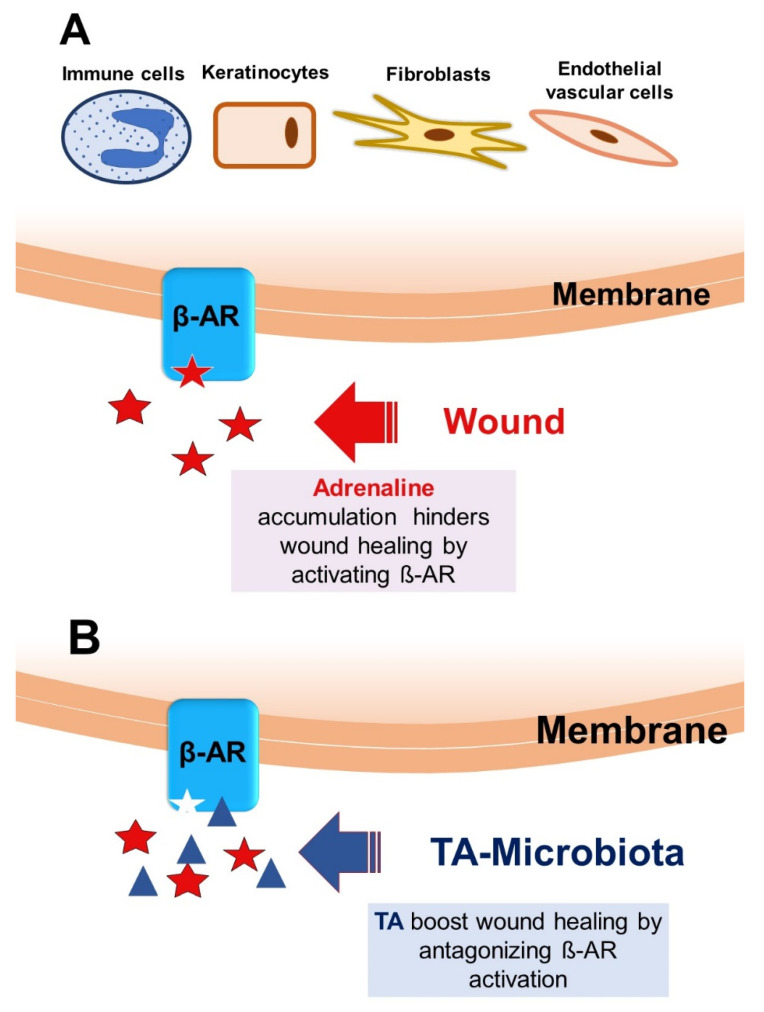
The role of adrenaline and TA in wound healing. (**A**) Cells of the epidermis, dermis, and hypodermis express ß-AR and are therefore responsive to elevated adrenaline levels triggered by wounding as a stress reaction. Sustained ß-AR activation due to elevated adrenaline levels results in a number of sequelae: (a) impairment of the migration ability of keratinocytes and re-epithelization; (b) delay of infiltration of the immune cells to the wound, as well as delay of cytokine production and macrophage recruitment; (c) decrease in collagen production and migration of fibroblasts; (d) inhibition of angiogenesis by endothelial vascular cells. (**B**) TAs, particularly those produced by skin microbiota, antagonize the effect of adrenaline by interacting allosterically with ß2-AR, thus boosting wound closing.

**Figure 2 ijms-22-04996-f002:**
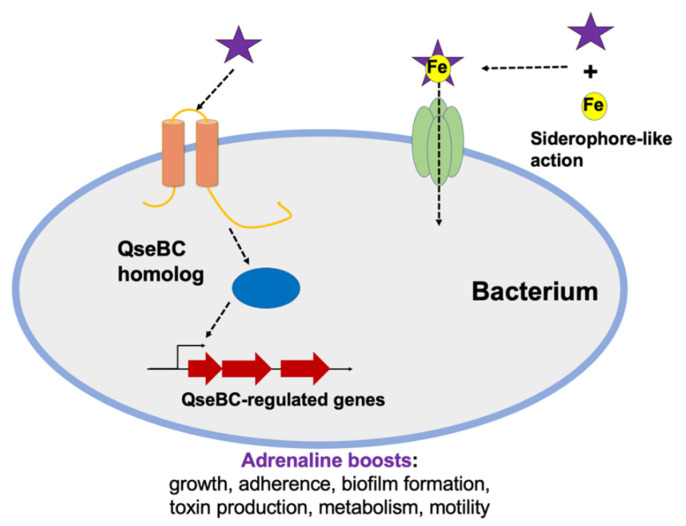
Adrenaline affects skin bacteria. The adrenaline level by injury stress also affects some skin bacteria possessing QseBC-regulated pathways. Adrenaline acts as a QseBC quorum-sensing activator, thus inducing bacterial motility, biofilm formation, adherence, and toxin production via this signaling pathway. In addition, adrenaline may also support bacterial growth and metabolism by complexing ferric ions as a siderophore, thus supplying these bacteria with additional iron.

**Figure 3 ijms-22-04996-f003:**
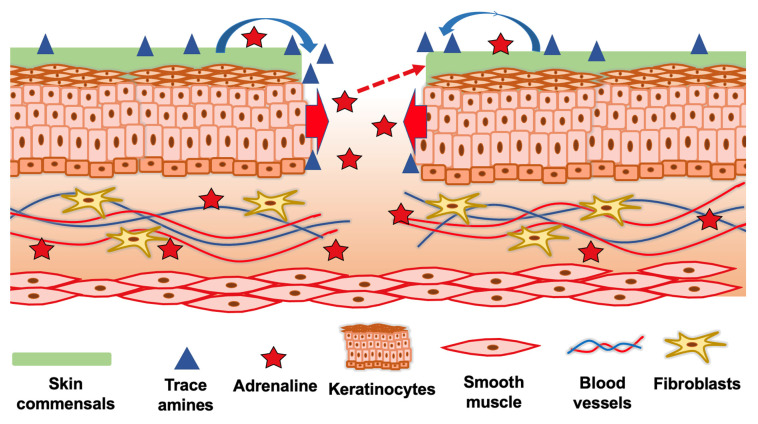
Interplay between trace amines (TAs) and adrenaline in wound healing. An injured skin tissue triggers an increased adrenaline level locally and systemically. The prolonged adrenaline exposure in the wound might hinder the healing process, mainly due to β2-AR activation. Some skin commensals are able to produce TAs (blue bent arrow), which act as β2-AR partial antagonists and help to accelerate the wound healing. On the other hand, adrenaline is also able to activate QseBC-regulated pathways and thus promote virulence traits of some members of the skin microbiota (red dashed arrow) that may delay wound healing.

**Table 1 ijms-22-04996-t001:** Microbiota species that have an adverse (A) or a promoting (B) effect on wound healing.

Species	Effector	Mode of Action	References
**(A) Adverse effect on wound healing**			
*Peptostreptococcus* spp. (*P. magnus, P. vaginalis,* and *P. asaccharolyticus*)	Bacterial supernatant	Inhibit fibroblast proliferation, keratinocyte proliferation and repopulation, and endothelial tubule formation	[17]
*Staphylococcus aureus, Pseudomonas aeruginosa, Peptoniphilus* sp., *Stenotrophomonas* sp., *Finegoldia* sp., *Serratia* sp., *Bacillus* sp., *Enterococcus* sp., *Enterobacter aerogenes, Acinetobacter baumanii, Klebsiella pneumoniae, Proteus mirabilis, Aspergillus fumigatus,* *Enterobacter cloacae, Corynebacterium frankenforstense*, *Corynebacterium striatum,* *Alcaligenes faecalis* and *Acinetobacter* sp.	Biofilm	Related to wound chronicity	[12,13,39,40]
Skin commensals	Can be particulate cell wall peptidoglycan	Augment *S. aureus* pathogenesis	[41]
Bacteria	Muramyl dipeptide	Delay wound repair by reducing re-epithelialization; increasing inflammation; and upregulating of murine β-defensins 1, 3, and 14	[42]
Pathogenic staphylococci	Secreted factors	Activate the mitogen-activated protein kinase and phosphatidylinositol 3-kinase/AKT signaling pathways and suppress NF-κB activation	[28]
Pathogenic fungi		Wound necrosis	[43]
**(B) Promoting effect on wound healing**			
*Staphylococcus epidermidis* and possibly other skin commensals with *sadA* gene	Trace amines	Accelerate wound healing by partially antagonizing the β-adrenergic receptor	[19,44]
Skin commensals predominantly from *Staphylococcus*	Bacteriocins	Inhibit pathogenic Gram-positive bacteria such as *Cutibacterium acnes*, *Staphylococcus epidermidis,* and MRSA	[45]
*Staphylococcus caprae* and other coagulase-negative staphylococci	Autoinducing peptide	Inhibit quorum sensing of *S. aureus*	[31,46]
Lactobacilli	Organic acids	Antimicrobial activity against skin pathogens and prevent biofilm formation	[37,47,48]
*Lactobacillus rhamnosus, Lactobacillus reuteri*	Lysate	Increase keratinocyte proliferation and migration	[49]
*S. epidermidis*	Short chain fatty acids	Suppress the growth of *S. aureus* and *C. acnes*	[50]
*S. epidermidis*	Delta-toxin (PSMγ)	Cooperates with the host-derived antimicrobial peptides in the innate immune system to eliminate pathogens	[51]
*S. epidermidis*		Induces IL-17A+ CD8+ T cells, enhances innate barrier immunity, and limits pathogen invasion	[9]
*S. epidermidis* and *S. hominis*	Antimicrobial peptides	Selectively kill *S. aureus* and synergize with the human AMP LL-37	[36]
Commensal staphylococci	Secreted factors	Induce expression of the AMPs HBD-3 and RNase7 in primary human keratinocytes via Toll-like receptor (TLR)-2, EGFR, and NF-κB activation	[28]
Commensal staphylococci	Lipoteichoic acid	Inhibit both inflammatory cytokine release from keratinocytes and inflammation triggered by injury through a TLR2-dependent mechanism	[52]
*S. aureus*	Peptidoglycan	Ameliorate cyclophosphamide-impaired wound healing	[53]
Staphylococci	Surface proteins SpA and Sbi	Initiate signaling cascades that lead to the early recruitment of neutrophils, modulate their lifespan in the skin milieu, and contribute to proper abscess formation and bacterial eradication	[30]
Skin commensals		Trigger activation of neutrophils to express the chemokine CXCL10 to kill exposed microbiota; activate pDC to produce type I IFNs, which accelerate wound closure by triggering skin inflammation and early T-cell-independent wound repair responses	[29]
Skin commensals		Induce T-cell responses that lead to protection from pathogens and accelerated skin wound closure	[54]

## Data Availability

Not applicable.

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
