# Peer review of "The Ambivalent Role of Skin Microbiota and Adrenaline in Wound Healing and the Interplay between Them"

_ijms, 2021, doi:10.3390/ijms22094996_

Round 1

Reviewer 1 Report

The authors have produced a very long and somewhat rambling review that focuses on the role of microbiota, through signaling through the adrenergic receptor, might have an influence on wound healing. The paper is way too long and talks about many superfluous topics. The paper should be shortened to focus on the topic of microbiota and its influence on wound healing. For instance, the readers do not need to have a review on the basic phases of wound healing (at the beginning). If the paper is shortened and focused, I would consider the paper for publication. Otherwise, I would reject it.

Author Response

We thank the reviewer for the comment. We have shortened the paper significantly by deleting some parts which are considered to be less-related to the title and the main message.

  • We drastically shortened the basic phases of wound healing and
  • We deleted the immune stimulation part entitled "Excessive induction of the immune system impairs wound healing". Although we believe that this is a main reason for impairing wound healing.

We also changed the title to: “The ambivalent role of skin microbiota and adrenaline in wound healing and the interplay between them”

Reviewer 2 Report

This is an interesting manuscript and the authors have collected information useful to understand specific signaling involvement regarding wound healing and skin microbiota.

 The paper is generally well written and structured. However, in my opinion the paper has some shortcomings in regards to some aspects, and I feel these  concepts have not been utilized to its full extent. In several instances I suggest to cite more relevant and recent literature.

Furthermore I make additional suggestions for more in-depth analyses of the data.

Key critical points are

  1. a) the development of the signaling pathway to ARs relationship to wound healing in order to investigate possible therapeutic strategies
  2. b) a lack of a discussion of the possible receptor targets to analyze the impact of infections on care of patients affected by chronic wounds.
  3. c) the authors could be focused their attention on the study of specific signaling pathway possibly responsible of the worsening of the wounds

Author Response

We thank the reviewer for the comments and suggestions.

1. We added a short section in the paper that discuss about the possible therapeutics strategis related to our topic.

Topical probiotics are a therapeutic option in the treatment of chronic wounds. Wound healing is a complex process and orchestrated by sophisticated interplay between factors. As one of the intrinsic factors is adrenaline which is crucial for the early stages of the wound healing process via ARs activation. The extended ARs activation, however, impedes the keratinocytes migration, decrease collagen production, delays the immune cells activities and angiogenesis process. To overcome these impediments by ARs, the administration and application of ARs antagonists, particularly β-AR antagonist, showed promising results both in in vitro and in vivo studies. Studies by Romana-Souza (Romana-Souza et al., 2010a, Romana-Souza et al., 2010b) in stressed mice revealed that the administration of propranolol, a β-AR antagonist, reversed the delay in infiltration of neutrophils and mast cells into the wound area, a delay in TNF-α expression, and recruitment of F4/80-positive macrophages which later followed by faster wound closing and re-epithelization. Besides propranolol, dopamine and TA application could increase the keratinocytes migration rate by inhibiting the increase of cAMP level through β2-AR partial blockade and eventually accelerate wound healing (Luqman et al., 2020a).

Another potential therapeutic strategy to enchance wound healing is via skin microbiota manipulation. The purpose of skin microbiota manipulation is to increase the number of  advantageous microbiota, particularly in the wound healing process, either by preventing pathogen infection or accelerating the healing process (Table 1). As mentioned above TA can be produced by skin commensals that possess sadA gene. Indeed, it has been already demonstrated that topical application of TA-producing bacteria (S. epidermidis O47) on skin could enhance wound healing (Luqman et al., 2020a). Even certain lactobacilli, which are not typical skin commensals, and their lysate can increase keratinocytes migration and proliferation (Mohammedsaeed et al., 2015).  Moreover, Lactobacilli are known to produce organic acids that show antimicrobial activity against skin pathogen and inhibit biofilm formation on wound (Fijan et al., 2019, Lopes et al., 2017, Valdez et al., 2005). Skin microbiota manipulation can be also carried out by topical application of certain compounds, such as traditional Chinese medicine that has been proven to be effective in treating chronic ulcer by regulating wound microbiota (Wu et al., 2018).

Conclusion

Skin microbiota and AR play an ambivalent role in wound healing. On the one hand, it can hinder the wound healing process up to chronic wound, on the other hand, some bacterial species can accelerate the wound healing process and suppress the colonization of pathogens. Skin injury induces the stress hormone adrenaline, which delays early stage wound healing by activating ß-AR in skin cells. However, adrenaline not only affects skin cells it also promotes virulence and growth of unwanted bacteria. TA producing skin commensals can override the effect of adrenaline and thus positively influence wound healing (Fig. 3). These bacteria are part of the skin microbiota. However, their proportion varies from person to person and may be so small for some that the positive effect does not come into play. Therefore, selected TA-producing commensal bacteria may represent a promising therapeutic option in wound healing.  

2 and 3. As we focus on the adrenergic receptors (ARs), we discussed about how the ARs play roles in orchestrating wound healing, either hindering or enhancing the process. We also discuss about the specific pathways regarding the role of ARs in hindering and enhancing the wound healing. As we re-arranged the paper to make it more relate with the new title, we hope that the role of AR and the related specific pathways can be highlighted and understood better.

Round 2

Reviewer 1 Report

Accept

This manuscript is a resubmission of an earlier submission. The following is a list of the peer review reports and author responses from that submission.